# Ammonia Detection by Electronic Noses for a Safer Work Environment

**DOI:** 10.3390/s24103152

**Published:** 2024-05-15

**Authors:** Tiago Reis, Pedro Catalão Moura, Débora Gonçalves, Paulo A. Ribeiro, Valentina Vassilenko, Maria Helena Fino, Maria Raposo

**Affiliations:** 1Laboratory of Instrumentation, Biomedical Engineering and Radiation Physics (LIBPhys-UNL), Department of Physics, NOVA School of Science and Technology, NOVA University of Lisbon, 2829-516 Caparica, Portugal; tff.reis@campus.fct.unl.pt (T.R.); pr.moura@campus.fct.unl.pt (P.C.M.); pfr@fct.unl.pt (P.A.R.); vv@fct.unl.pt (V.V.); 2Institute of Physics of Sao Carlos, University of Sao Paulo, São Carlos 13566-590, Brazil; gdebora@ifsc.usp.br; 3LASI—Associated Laboratory of Intelligent Systems, CTS—Centre for Technology and Systems, UNINOVA, Department of Electrotechnical and Computer Engineering, NOVA School of Science and Technology, NOVA University of Lisbon, 2829-516 Caparica, Portugal; hfino@fct.unl.pt

**Keywords:** ammonia, e-nose, environment, health at work, review

## Abstract

Providing employees with proper work conditions should be one of the main concerns of any employer. Even so, in many cases, work shifts chronically expose the workers to a wide range of potentially harmful compounds, such as ammonia. Ammonia has been present in the composition of products commonly used in a wide range of industries, namely production in lines, and also laboratories, schools, hospitals, and others. Chronic exposure to ammonia can yield several diseases, such as irritation and pruritus, as well as inflammation of ocular, cutaneous, and respiratory tissues. In more extreme cases, exposure to ammonia is also related to dyspnea, progressive cyanosis, and pulmonary edema. As such, the use of ammonia needs to be properly regulated and monitored to ensure safer work environments. The Occupational Safety and Health Administration and the European Agency for Safety and Health at Work have already commissioned regulations on the acceptable limits of exposure to ammonia. Nevertheless, the monitoring of ammonia gas is still not normalized because appropriate sensors can be difficult to find as commercially available products. To help promote promising methods of developing ammonia sensors, this work will compile and compare the results published so far.

## 1. Introduction

Ammonia has been used in a wide range of applications, including industrial production of fertilizers and explosives, as well as in medical diagnosis, in which ammonia can act as a biomarker for several health conditions [1,2,3,4,5,6,7,8,9,10]. However, ammonia gas is also a hazardous substance and severely irritating to human eyes, skin, and membranes [11,12,13]. In fact, high concentrations of ammonia have been related to blindness, burns, dyspnea, progressive cyanosis, and even death from either suffocation or pulmonary edema [14].

Ammonia is a cornerstone in fertilizers and medicine and can be found in dyes, explosives, and resins [7]. Ammonia is the best source of chemicals with nitrogen [5,6]. However, ammonia used in agriculture contaminates groundwater, and ammonia pollution is of such a concern that research in ammonia stripping, an area that seeks methods to neutralize pollution, has accumulated much interest [4,8]. Additionally, ammonia can be found in many commercial cleaning products; in this way, it can reach places unrelated to agriculture [2,3]. This is the biggest concern in this work, as it introduces the public to ammonia concentrations greater than the established permissible exposure limits (PELs). Several institutes around the world have targeted ammonia as a compound of interest and have already set exposure limits to it. The current PELs, endorsed by the Occupational Safety and Health Administration (OSHA), have been keeping the same values since 1992, and are referred to in [14,15]. The proposed PELs values range from 25 to 35 ppm_v_, and the most relevant ones are 25 ppm_v_ for an 8 h time-weighted average (TWA) and 35 ppm_v_ as a short-term exposure limit (STEL), i.e., 15 min TWA. Later, on the 8 June 2000, the European Agency for Safety and Health at Work commissioned more restrictive exposure limits for ammonia. The PEL was set to 20 ppm_v_ for 8 h TWA daily exposure and 50 ppm_v_ for the STEL [16].

Despite all studies already established and content previously available, the monitoring of ammonia gas is still not normalized mainly due to the unavailability of appropriate commercial sensors. A lack of standard procedures used to construct ammonia sensors is a major contemporary challenge in assessing ammonia. Additionally, there is a lack of appropriate sensors used to detect ammonia, evaluate work conditions for all kinds of workplaces, and enable accurate identification of main health impacts on human organisms in chronic exposure scenarios. Our review can help to promote the most promising methods for the development of ammonia sensors and update options in ammonia detection aimed at determining accurate detection levels to satisfy previously mentioned regulations, i.e., up to 20 ppm_v_ [16].

Several sensors for the detection of ammonia have been produced and some of them have the sensitivity to detect in the concentration range referred to above. But, the range of measurement, as well as the limits of detection, can be reduced if, instead of using a unique sensor of ammonia, an array of sensors is used and data are treated with supervised and unsupervised mathematical methods, now designated as artificial intelligence, as demonstrated in Ref. [17]. This approach also allows for the qualitative and quantitative detection of molecules in complex media, which is the case of the environment in which many molecules other than ammonia are present. These kinds of sensing devices are designated in the literature as electronic tongues or electronic noses (e-noses) if the samples to be analyzed are liquid or gas, respectively. The literature shows that several e-noses have been developed and some are associated with ammonia detection.

To highlight the relevancy of the detection of ammonia in the environment, we searched the scientific research published annually from 2010 until now under the scope of this topic. The results were gathered from one of the main indexing platforms, Web of Science, and the keywords used in bibliographic databases were “ammonia AND environment”, “ammonia AND e-nose”, and “ammonia AND e-nose AND environment”. E-noses were initially chosen as an identifier of ammonia sensors indicating how relevant they have become. To compare the results, the ratios between the number of publications on “ammonia AND e-nose AND environment” relative to the number of publications on “ammonia AND e-nose” per year as well as the number of publications on “ammonia AND e-nose AND environment” per year were plotted in Figure 1. From this figure, it can be concluded that the topic “ammonia AND e-nose AND environment” represents a set of 20% of the topic “ammonia AND e-nose”, albeit with very high variance, and that, as such, it has also seen an increase in relevancy since 2010. This suggests that research in ammonia has merged both contexts for the environment and e-nose and that it is relevant and in demand nowadays. The importance of detection is also demonstrated by the fact that ammonia is one of the volatile organic compounds (VOCs) expelled in breath and is associated with diseases. This topic is also covered in this article.

This work addresses the development of sensors for the assessment of ammonia as a contribution to results already described in this field. To achieve this, it provides a background by introducing ammonia, then explains the overall working principle of electronics, compares the most important sensors in the field, evaluates their impact on air quality control and working conditions, demonstrates the effectiveness of using sensor arrays that can work as an e-nose and, finally, describes other applications of e-noses designed for ammonia detection. 

## 2. Sensors of Ammonia

Commercially miniaturized photoionization sensors (mini-PID sensors) are produced by Honeywell [18], Industrial Scientific [19,20], and ION [21,22] to detect ammonia in the context of safety in the work environment. PID technology is portable and offers real-time analysis at a reasonable price [23]; however, it lacks selectivity [24]. As such, it is necessary to look for alternatives that provide the same benefits and high selectivity.

A prior review of ammonia sensors has shown examples of different types of sensors that have been developed [25], and all of them are indicated in Figure 2. To infer about the sensing mechanisms associated with these sensors, one can summarize the following:

**Metal Oxides:** There are two types of metal oxides, n-type and p-type, the n-type being more sensitive to ammonia. Depending on the material used in the electronic structures, metal oxides are either non-transition metal oxides or transition metal oxides. Common pre-transition metal oxides are MgO and Al_2_O_3_, while common post-transition metal oxides are ZnO and SnO_2_. Experimental results demonstrate that metal oxides need to have in their electronic configuration orbitals d unfilled (d0) or preferably filled (d10) to detect ammonia.

**Conducting Polymers:** Most conducting polymer sensors utilize the redox reaction between the target gas and the polymer to detect it. Polymers have a high surface-to-volume ratio. This improves adsorption/desorption, resulting in high sensitivity and quick response and recovery times.

**Diode Laser Absorption Spectroscopy (TDLAS):** This is an optical sensor that uses infrared light to detect target gases. The wavelength range where NH_3_ absorbs light is between 1450 and 1560 nm, and as such it can be detected by TDLAS. For this, quantum cascade lasers (QCLs) are used due to their high resolution, operating at RT, sensitivity, and selectivity. Having a narrow linewidth and being single longitudinal are critical for QCL spectroscopy; as such, a distributed feedback (DFB) structure is used at a selected wavelength.

**Electrochemical:** These sensors consist of three electrodes (sensing, counter, and reference electrodes) immersed in an electrolyte. Solid-state electrolyte sensors use amperometry and potentiometric techniques and as such suffer from electrolyte poisoning and high operation temperature (>200 °C). Liquid-state electrolyte sensors use voltametric and potential step chronoamperometry and as such suffer from a low lifespan due to the degradation of the electrolyte. Room-temperature ionic liquids (RTILs), derived from ionic liquids (ILs), have been used to replace the electrolytes, and results show durability up to 5 V, liquid at RT, high conductivity, thermal stability, and polarity. However, they also have low diffusion coefficients and, as a result, small current signals

**Surface Acoustic Wave (SAW):** These sensors consist of two interdigitated transducers, made of noble metals such as gold (IDTs), on a substrate with a delay line between them. SAW analysis of the surface perturbations in the delay line show that they are caused by the gas.

**Field-Effect Transistor (FET):** These sensors adopt the basic structure of a metal insulator–semiconductor (MIS) junction and measure the change in the current–voltage (I-V). A metal–oxide–semiconductor field-effect transistor (MOSFET) model is used, where the gate holds the film (typically catalytic metals). Platinum, palladium, iridium, or mixtures of those metals are used for NH3 detection. Thickness, structure, or morphology play an important role in sensitivity. FET sensors have low power consumption in ambient conditions and cost-efficient production. However, they also have high operation temperatures (100~300 °C). Organic field-effect transistors (OFETs) have advantages compared to FETs, such as flexibility, sensitivity, low cost, weight, and ease of production. P-type metal oxides have higher hole mobility than n-type metal oxides directly, improving their performance.

Analyzing the properties of these sensors described in the literature for ammonia detection, one can conclude that metal oxide sensors utilize n-type and p-type oxides, with n-type being more sensitive. Conducting polymer sensors rely on redox reactions and offer high sensitivity and quick response times due to their high surface-to-volume ratio. TDLAS sensors use infrared light to detect gases like NH_3_. Electrochemical sensors employ different electrode configurations and suffer from electrolyte-related issues, with room-temperature ionic liquids showing promise as replacements. SAW sensors analyze surface perturbations caused by gas. FET sensors measure current–voltage changes in noble metals to detect the target gas and have variations such as MOSFET and OFET. However, all these different methods for sensing ammonia have advantages and disadvantages. The advantages of metal oxides are their simplicity, low cost, and flexibility in fabrication [26,27], and the disadvantages are their high operation temperature and low sensitivity [28]. The advantages of conducting polymers are their low limit of detection (LoD) and quick response time [29,30,31], and their disadvantage is low mechanical strength [32]. The advantages of TDLAS are its simplicity of operation and high selectivity, rapidness, great sensitivity, long lifecycle, and lack of measurement error from the long-term memory effect [33,34], and its disadvantages are the high currents and cryogenic temperatures required [35]. The advantages of the most common electrochemical method are its lower power consumption and cost efficiency, and the disadvantage is its low selectivity [36]. The advantages of SAWs are their fast response time, sensitivity, and stability even in harsh environments [37]. The advantages of FETs are that they are small and cheap sensors, and their disadvantage is the high operation temperature [38]. However, workarounds have been developed to overcome the shortcomings of most of these types [25]. The schematic configurations of the detection of each type of sensor are illustrated in Figure 3.

Suitable sensors for ensuring work safety must accurately detect NH_3_ levels up to 20 ppm_v_ within a short period, operate at room temperature, and be portable to be practical. Among sensing methods, TDLAS and electrochemistry are challenging to integrate into portable devices; therefore, they are beyond the scope of this paper. However, it is worth noting that Alphasense has developed portable electrochemical NH_3_ sensors [41,42], as well as optical particle counters [43,44]. Additionally, most technologies based on FETs and conducting polymers usually exhibit slow response times as a tradeoff in favor of achieving concentrations in parts per billion (ppb_v_), which is contrary to what is desirable. Metal oxides also require high operating temperatures, thus suggesting that none of the mentioned technologies emerge as the best option for implementing efficient sensors with specifications compared on a case-by-case basis. The collected literature that presents unique sensors and their respective results is [1,26,27,30,31,33,34,35,37,38,39,40,45,46,47,48,49,50,51,52,53,54,55,56,57,58,59,60,61,62,63,64,65,66,67,68,69,70,71,72,73,74,75,76,77,78] and, from these, Table 1 shows characteristics of sensors with a response time under 1.3 min at a concentration in the range of 5 to 50 ppm_v_ of NH_3_ operating at RT, a portable technology, and 20 ppm_v_ within the range of concentrations tested. The parameters associated with the sensor performance presented in Table 1 were sensitivity, response time, and recovery time. These parameters are defined as follows: (1) sensitivity is the slope of the output characteristic curve or, more generally, the minimum input of physical parameters that creates a detectable output change; (2) response time is the time it takes for the sensor to change its output signal from the initial state to a certain percentage, commonly 90%, of the final value; and (3) recovery time is the time taken for a sensor to return to its baseline value after the step removal of the measured variable. It should be noted that selectivity, the ability to discriminate the target from the interference molecules and display a target-specific sensor response, is not indicated in the table since the presented sensors have some selectivity but, certainly, this characteristic was not completely tested in the listed sensors in the table. The values listed in Table 1 were chosen from those available by each work closest to 20 ppm_v_.

To select the best sensors among those already mentioned, the criterion for a suitable response time was narrowed from 1.3 min to 10 s as it makes it far more practical to use. Only five sensors are within the range of this adjustment, and they can be either based on metal oxides or polymers. Three out of these five sensors describe the use of polyaniline (PANI) for sensing NH_3_, and they have been obtained in different groups, indicating that PANI is the best option to be used in the context of safety in the work environment. Between these three, the sensor produced by Tai et al. [59] has shown by far the best sensibility and response time, and it can be appointed as the best sensor briefly. However, it is worth looking into the methods employed by all these three in detail as all these works had interesting results. The film was produced with the common in situ self-assembly technique, where a positive surface was created with PDDA/PSS before thin PANI/TiO2 films were deposited under different polymerization temperatures [59]. The most distinguished part of the process was controlling temperature during the preparation of PANI/TiO_2_ film. This sensor showed the fastest response time, and highest sensitivities, and, finally, was one of the few with results on the lifecycle. PANI nanofibers doped with TSA were synthesized by a dilute polymerization method and allowed one to obtain a greater sensing performance than those obtained with conventional PANI, which has been attributed to a large specific surface area and interconnected network structures [48]. Thin PANI film was intercalated with Cu nanoparticles, and sensor films containing 0.13% Cu showed the best results [35], where adding Cu to a PANI film improved the response and response time. Additionally, the process used in [58,60] should also be referred to as they also handled PANI/TiO_2_ films. The process used by [60] was like the one shown in [59], where the main difference was that the temperature when producing the thin films was not controlled. The description also does not suggest that the PDDA/PPS layer was not removed, and as such the films yielded the worst results (nevertheless, these results are still better than others). It is worth noting that this sensor shows the best sensitivity among the PANI sensors. In [58], PANI was processed by the chemical oxidative polymerization method while TiO_2_ was synthesized by the sol–gel method, with thin film being assembled on glass substrates by spin coating at 3000 rpm for 40 s and dried on a hot plate at 100 °C. An interesting result from this study was the high selectivity exhibited by thin PANI/TiO_2_ films. Further research that tests combining these technologies and verifies the results could be valuable in creating a more robust sensor. However, as the production of suitable sensors becomes more complex by different protocols, the sensor also becomes more expensive, making it less competitive. As such, it is important to highlight the most relevant results from differences appointed in the methodologies.

There were many other sensors in the collected literature with promising results; however, some sacrificed response time for a higher limit of detection (LoD), and others did not test a range of concentrations that included the 20 ppm_v_ mark, as they were developed with different purposes in mind. Another issue found was the lack of tables and comparative data; most research shows an image and derives either relative or qualitative conclusions, making it harder to objectively compare research. This is why some values in the tables are not exact but approximations based on the illustrated response in the graphs. Additionally, the lack of a universal set of variables to characterize the sensor makes it impossible to conduct a methodical review. Therefore, through this research, we propose that, when presenting data on a novel sensor, the research includes a table with all the relevant values, namely response time, recovery time, sensitivity, R^2^, operating temperature, LoD, detection range, and type. Regarding sensitivity, there was a detachment between absolute, relative, and normalized values at specific concentrations. This problem can be observed in the table, where a wide range of units can be seen in a small number of considered literature research, and, to assist in solving this issue, we suggest that the normalized response without % can be used. Additionally, sensitivity has been referred to as a slope between response and concentration rather than a response to a certain concentration. Most researchers already consider this when determining the best range of concentration for sensors to operate in and choose a range where a significant variation in concentrations can be observed. However, with the current method for considering the response as sensitivity, even a sensor that can distinguish different concentrations would be considered good if the response is high. On a similar note, few researchers valued calibration, and most did not present a fitting. However, there are free and easily accessible tools that automatically calculate a linear regression for a dataset and present the sensitivity, as well as the R^2^, which represents how accurately the fitting matches the data. Regarding the LoD, some researchers consider it the lowest concentration measured while others do not consider it entirely; however, that is not its definition, and it should instead be properly calculated as 3.3×σ/sensitivity, where σ is the standard deviation. Finally, every sensor should clearly state which of the previously mentioned types it belongs to, as it can be ambiguous in some cases.

**Table 1 sensors-24-03152-t001:** Ammonia sensors with a response time under 1.3 min at a concentration in the range of 5 to 50 ppm_v_. C is concentration.

Sensor Type	Film Type	Detection Range [ppm]		Sensitivity			Response Time		Recovery Time		References
		Min	Max	Value	Units	C [ppm]	Value [s]	C [ppm]	Value [s]	C [ppm]	
metal oxide	SnO_2_/In_2_O_3_	0.1	10,000	53	(relative R)	10	7	1	10	1	[55]
metal oxide	TiO_2_	5	100	220	(relative G)	20	34	5	90	5	[57]
metal oxide	PANI/TiO_2_	20	100	10	[%] (normalized R)	20	~63	20	~41	20	[58]
metal oxide	PANI/TiO_2_	0	150	1.3	(normalized R)	23	2	23	60	23	[59]
metal oxide	PANI/TiO_2_	0	141	1.67	(normalized R)	23	18	23	58	23	[60]
metal oxide	Sr/SnO_2_	10	2000	~15	[%] (normalized R)	20	16	10	-		[61]
metal oxide	NiO/ZnO	15	75	30	[%] (normalized R)	30	27	50	150	50	[62]
metal oxide	MoS_2_/ZnO	2.5	100	~37	[%] (normalized R)	10	10	50	11	50	[47]
metal oxide	MoS_2_/PDDA	2.5	100	~27	[%] (normalized R)	10	7	50	11	50	[47]
conducting polymer	PANI	5	200	27	[%] (normalized R)	5	10	50	100	50	[48]
conducting polymer	Cu-PANI	1	100	~86	[%] (normalized I)	50	7	50	160	50	[49]
conducting polymer	graphene/PANI	1	6400	3.65	[%] (normalized R)	20	50	100	23	100	[50]
conducting polymer	B-N unit	1	80	300	(relative R)	20	65	40	25	40	[75]
conducting polymer	CPNW	0.8	59	~18	(normalized R)	23	~10	23	~200	23	[31]
SAW	ZnO/SiO_2_	5	120	2000	[Hz]	30	40	30	28	30	[51]
SAW	Co_3_O_4_/SiO_2_	1	60	4000	[Hz]	20	~80	20	~110	20	[53]
SAW	PANI	20	70	1.79	ppm [Hz]	20.45	~80	20	~75	20	[40]
FET	PQT-12	10	80	23.8	[%] (normalized I)	20	~45	80	~80	80	[54]

## 3. Operation and Characterization of e-Noses

All the results and circumstances described above suggest that the best solution for achieving better results in ammonia detection is through the development of e-noses. This solution is also defended by [79]. An e-nose consists of an array of specific or non-specific sensors used to mimic human senses—see [80] and references therein—and then distinguish VOCs from the electrical signals, such as resistance, capacitance, and impedance, among others, which can be measured using changes in concentration that take place because of interactions of a sensing material with a gas sample, as illustrated in Figure 4. The literature described above reveals that adequate molecules/layers can detect ammonia molecules. Therefore, to detect ammonia, one should prepare sensors with sensing layers consisting of oxides (Ca-doped ZnO nanoparticles [39], indium oxide (In_2_O_3_:La) [77], Mo-doped (doped at different concentrations) Bi-VO4 [76], zinc oxide and cobalt-doped ZnO [81]); polymers (polyaniline (PANI) [48,49], P-BNT [75]; poly[benzimidazole-benzo-phenanthroline] (n-BBL) [74]); iron chalcogenide [82]; Prussian blue analogs (PBAs) [73]; 4-phenyl-1,8-naphthalimide (NMI) [83]; WS_2_ nanosheets [72]; and blends as ZnO/rGO [39], among others. The scheme shown in Figure 5 summarizes the materials adequate for detecting ammonia and which could be used in the development of e-noses.

Furthermore, e-noses should be built with arrays of sensors patterned to control changes in the relative humidity of a target gas and projected in such a way as to avoid signals of interference gases. Therefore, the array sensors can be used in a new generation of sensors for detecting different molecules in complex media that can be preferentially detected rapidly and non-invasively. Efficient e-noses should have some characteristics that include portability, selectivity, and sensitivity, added to a fast response and recovery time under different environmental conditions. For practical purposes, e-noses can be composed of different parts: a gas sampling pump, gas chamber, sensor array created with circuit boards, and data acquisition and signal recording systems. The data processing unit is one of the most important components of e-noses when searching for the best performance since one should use well-calibrated data libraries to compare the measured data and thus qualitatively and quantitatively detect the target molecule. The data processing can be carried out using machine learning algorithms, which include sample selection, data labeling, validation of the dataset, classification, and modeling [84,85,86]. One of the drawbacks of sensors and also of e-noses is inaccuracy due to the presence of interference gases or gases adsorbed on the sensors, showing a significant impact on the recovery times of most gas sensors. This drawback can be solved if the array of sensors is subjected to a temperature gradient or incorporated into a small vacuum chamber to remove the adsorbed molecules, and the results can become accurate. This surely increases the price of the sensing device. In addition, this methodology is aligned with sensors based on metal oxides and conducting polymers but can also decrease the response time of the e-nose.

## 4. Additional Applications for Ammonia e-Noses

As previously mentioned, ammonia can represent a risk to the health of chronically exposed beings, especially in work locations rich in emitting sources. Despite its toxicity and hazardousness, ammonia can also be a valuable source of information regarding the health condition of human organisms. Several e-noses have been developed aiming to detect ammonia in biological samples and consequently diagnose specific pathologies and health conditions [87,88,89].

Ammonia is endogenously present in the human organism and is involved in several biological processes mainly linked to food processing and waste disposal [90]. Urea is the result of the liver processing the toxic compounds often ingested during feeding and reducing them to less toxic soluble elements that can be excreted. These toxic elements commonly present in the bloodstream are processed in the liver, leading to the production of urea through a process vulgarly known as the urea cycle. Ammonia is often combined with carbon monoxide in the liver to form product elements that are necessary to the urea cycle, namely carbamoyl phosphate. Once concluded, the urea and all the waste elements, ammonia included, are absorbed into the bloodstream for consequent filtration in the kidneys. Finally, the waste is disposed to the exterior of the organism. If the liver exhibits limitations in performing the enzymatic processes mentioned above, or if the kidneys start losing their capability of filtering the blood, the levels of ammonia will be necessarily more elevated than normal and, consequently, several complications arise [88,91]. These complications can then be diagnosed and explored through the determination of the ammonia levels in blood and other biological fluids like urine or even exhaled air [87,92].

Considering all the mentioned facts, altered levels of ammonia in biological fluids can be related to impairment and dysfunction of the liver, kidneys, stomach, and even lungs. Among all the health conditions potentially traceable through the elevated levels of ammonia, the most relevant are chronic liver diseases (CLDs), chronic kidney diseases (CKDs), and the monitoring of hemodialysis treatment, cirrhosis, lung cancer, asthma, cystic fibrosis, and some others [10,88,93]. Within these specific purposes, e-noses have played a significant role in analyzing biological fluids, namely blood, urine, and breath, whose content of ammonia can represent an open window to the interior of the human body [87,89].

### 4.1. Hepatic Impairment

Elevated levels of ammonia in blood have been directly related to chronic liver diseases. Since these elevated levels are traceable in samples of biological fluids like urine or exhaled air, CLD has been deeply studied regarding the suitability of ammonia sensors for the development of methodologies that enable one to accurately, rapidly, and non-invasively diagnose pathologies [94,95].

Aiming to develop a system capable of monitoring ammonia levels in exhaled air in a real-time manner, Ishida et al. (2021) assembled a portable device containing a cuprous bromide sensor. Thin-film-sensor-based devices based on p-type semiconductor cuprous bromide were exposed to the exhaled air of 39 volunteers (21 CLD patients and 18 healthy subjects). The authors were able to successfully differentiate both groups of volunteers based on the considerably elevated levels of ammonia in the breath of CLD patients. In addition, the assembled e-nose could even prove that the significance values for the tests developed with blood and exhaled air samples were similar [96].

A prototypic device based on an array of semiconductor gas sensors was also used to monitor liver impairment through the levels of ammonia in the exhaled air. The e-nose consisted of an array with six metal–oxide–semiconductor gas sensors capable of detecting ammonia in complex matrices like the breath by producing an electrical resistance variation. A cohort of 64 volunteers, including 16 healthy subjects, 20 non-cirrhotic patients with CLD, 22 cirrhotic patients, and 6 cirrhotic patients with recent episodes of hepatic encephalopathy, was considered during the study. The authors were capable of successfully proving the suitability of the employed e-nose for purposes of monitoring liver impairment through ammonia in breath; nonetheless, the necessity of enhancing its performance was pointed out [97].

Cirrhosis, a short version of hepatic cirrhosis, is often a consequence of damages existent in the liver whose origin is directly related to conditions like hepatitis B or C or chronic use of alcoholic beverages. These lesions lead to the formation of scars that tend to replace healthy functional hepatic tissue and, consequently, to the impairment of the organ. Ammonia has been explored regarding its prognostic role in hepatic cirrhosis patients [98,99].

Being aware of the aforementioned facts, Spacek et al. (2015) employed an e-nose to study ammonia in the breath and differentiate among two groups (10 healthy volunteers and 10 cirrhotic patients) of a cohort. The authors were able to verify that ammonia levels in the breath of healthy subjects (379 pmol mL^−1^) are superior to the levels registered for cirrhotic patients (350 pmol mL^−1^), a relationship also verified in the levels of ammonia in blood [100].

Cirrhotic patients were equally the target in the study developed by Adrover et al. (2012). The authors gathered 106 subjects, specifically 55 cirrhotic patients and 51 healthy volunteers, who were later studied through their exhaled air. An e-nose was employed to perform the analyses. Its working principle was based on registering the voltage variation of an electronic sensor when exposed to ammonia. The authors could prove that ammonia is present in the breath of healthy people at an average value of 151.4 ppb_v_. This value increases to 162.9 ppb_v_ and 184.1 ppb_v_ in scenarios of cirrhosis and cirrhosis with hepatic encephalopathy [101].

A final example of the utility of e-noses in the assessment of liver impairment through the ammonia levels in breath is the work of Voss et al. (2022). Here, the authors developed and employed an e-nose system to assess the emissions of ammonia in the breath of cirrhotic patients. Metal–oxide–semiconductor gas sensors were used to assemble the sensor array of the prototype. Then, the device was exposed to the exhaled air of 30 volunteers, including 10 healthy individuals, 10 patients with compensated cirrhosis, and 1 with decompensated cirrhosis. The e-nose developed by the authors enabled them to differentiate between the considered groups with accuracy levels over 95%, proving the suitability of this type of system for both liver impairment monitoring and modern medicine overall [102].

The field of e-noses can play an important role in the assessment of ammonia in body fluids for purposes of monitoring chronic liver diseases, hepatic encephalopathy, and even cirrhosis. Nonetheless, it is important to mention that elevated levels of ammonia can be caused by several factors and not directly by the mentioned pathologies. Further studies and more accurate e-noses must be developed to fully characterize the relationship between ammonia and hepatic disorders [103,104].

### 4.2. Renal Impairment

Rapid and effective diagnosis procedures are particularly mandatory in patients suffering from chronic kidney disease (CKD) and other renal pathologies. This group of pathologies is responsible for a considerably high number of deaths every year and many more chronic comorbidities in patients, namely acute renal failures and the necessity of hemodialysis treatment for long periods of time. Due to the direct connection between ammonia and CKD mentioned above, this molecule can act as a biomarker to diagnose the pathology in a rapid, accurate, and low-cost manner [105,106].

Ricci et al. (2021), being aware of the potentialities of ammonia for diagnosing CKD, developed a system capable of detecting ammonia at the vapor phase. The system consisted of a thermodynamic sensor platform containing microheaters deposited onto ultrathin yttria-stabilized zirconia ribbons that, in turn, were coated with a metal oxide catalyst for purposes of detecting the target analyte. The working principle of this sensing technique consists of registering the changes in the electrical power required to maintain the sensor at a specific temperature when exposed to the ammonia molecule. This exposure leads to the decomposition of ammonia upon contact with the catalyst surface and the consequent release of the energy responsible for the alterations in the electrical power. The system developed by the authors proved to be capable of detecting ammonia at concentration levels as low as 5 ppm_v_, denoting an auspicious future for its application in medical scenarios [107].

Aiming to assess the levels of ammonia in the exhaled air of CKD stage-one-to-five patients, Chan et al. (2020) employed a vertical-channel organic semiconductor sensor-based system for sample screening. A cohort of 121 CKD patients (stage one: 19 patients, stage two: 26 patients, stage three: 38 patients, stage four: 21 patients, and stage five: 17 patients) was considered during the study. The samples of exhaled air were collected into 500 mL plastic bags. The working principle of the analysis system consisted of measuring the electrical signal of the sensor when exposed to the sample. This e-nose proved to be suitable for breath analysis since it allowed one to differentiate different-stage patients with levels of sensitivity and specificity ranging from 69 to 80% and 69 to 95%, respectively. Furthermore, it allowed one to accurately detect ammonia at concentration levels ranging from 636 to 12,781 ppb_v_ [9].

Knowing the sensing capacity of graphene-based sensors, Lee et al. (2021) developed an array of graphene oxide sensors dedicated to studying three specific volatile organic compounds known for their potential to act as CKD biomarkers: isopropanol, acetone, and ammonia. The authors prepared standard samples of the target compounds at defined concentrations and registered the variations in the electrical resistance of the sensors when exposed to the targets. Overall, the e-nose developed was able to identify compounds with accuracy levels above 90%. Additionally, the system could differentiate between healthy volunteers and CKD patients with accuracy levels of around 20% [108].

Severe cases of chronic kidney disease and other renal pathologies often result in the necessity of continued hemodialysis treatment. Ammonia has been studied regarding its suitability to act as a parameter to accurately assess the evolution of hemodialysis treatments in CKD patients. This suitability arises, as mentioned, from the deep connection between renal activity and the ammonia role in the urea cycle [109].

The work of Shahmoradi et al. (2021) is an example of the utility of e-noses in the field of ammonia detection for hemodialysis monitoring. The authors developed several arrays of sensors, specifically sulfonate-graphene-, graphene-oxide- and reduced graphene-oxide-based sensors and exposed them to samples of ammonia. These samples were previously developed aiming to mimic the concentrations of this molecule in the breath of renal impairment patients. Samples with concentrations ranging between 0.5 ppb_v_ and 12 ppm_v_ were considered for the study. The e-nose could successfully detect ammonia in all scenarios, proving its appropriateness for medical purposes [110].

Having a similar goal, Yabaş et al. (2023) developed an e-nose by synthesizing 4-pyridinyl-oxadiazole tetrasubstituted zinc and cobalt phthalocyanine compounds and mixing them with reduced graphene oxide to obtain hybrids, namely rGO, rGO/ZnPc, and rGO/CoPc. The authors studied a total of five volatile organic compounds, ammonia being one of them. Alterations in the electrical resistance of the sensors were registered during the exposure period to samples previously prepared, with concentrations ranging from 30 to 210 ppm_v_. As expected, the e-nose proved to be suitable for the goals of the study, namely, to detect different concentration levels of ammonia accurately and rapidly in volatile samples [111].

As visible, the field of e-noses for ammonia detection in biological samples for the diagnosis of chronic kidney disease renal pathologies overall has an auspicious future and there is no doubt that the medical field can rely on these newer technologies.

### 4.3. Gastric Impairment

As previously mentioned, the presence of ammonia in the human organism can be directly linked to the digestive and excretory systems. With this in mind, the assessment of ammonia concentration levels in body fluids and, specifically, in exhaled air, can be a valuable source of information regarding the condition of these systems. Peptic ulcers, halitosis, and even gastric cancer are examples of health conditions traceable through the detection of ammonia [87,88,112].

An example of the utility of e-noses in the detection of ammonia, as mentioned, is the work of Bouchikni et al. (2021). Here, the authors gathered a cohort composed of 8 gastric cancer patients and 13 healthy subjects to differentiate both groups solely based on their characteristic breath. To achieve this, breath samples were collected into inert Tedlar bags and immediately transferred to the e-nose. The e-nose was developed having as its base an array of SnO_2_ sensors whose response was registered during the exposure to the target samples. The authors could successfully distinguish both groups with a total explained variance of 98.33%, proving the suitability of the developed system for clinical purposes and, specifically, for monitoring gastric pathologies [113].

The presence of *Helicobacter pylori* in the stomach is often challenging to detect, the mucosal changes being the simplest and most accurate way. Besides its challenging detection, *H. pylori* can be responsible for several health conditions, like halitosis, gastric ulcers, inflammation of the stomach lining, and even gastric cancer. All these conditions can be mitigated if an accurate and rapid detection of the bacterium is achieved. Ammonia has been explored regarding its potential as a biomarker of the presence of *H. Pylori* [114,115].

Song et al. (2022) focused their work on developing novel e-noses for the assessment of the *H. pylori* presence. The developed system based its working principle on a fluorescent sensor, whose operation is known for being simple, allowing for good selectivity and a fast response. The authors reported this type of system as being the first one developed exclusively for ammonia detection in exhaled air samples and consequent diagnosis of *H. pillory* infection. In addition, the authors reported the detection of ammonia at very low levels of concentration, i.e., 22 ppb_v_, with response times of 80 ms [116].

Equally targeting the existence of H. pylori infection and consequent stomach cancer development, Zilberman et al. (2015) developed a system capable of detecting the presence of ammonia in the saliva of human subjects. The developed e-nose was built around a microfluidic optoelectronic sensor whose manufacturing consisted of embedding in a microfluidic array of microwells filled with ion-exchange polymer microbeads doped with several organic dyes. Once exposed to the target samples, the optical response of the sensors was monitored over a broad spectrum. The authors could successfully detect ammonia in saliva at the ppm_v_ levels of concentration, proving the suitability of the developed system for medical purposes [117].

It is evident how useful the field of e-noses for ammonia detection can be for medical purposes and, specifically, for the diagnosis of gastric diseases. The medicine of the future can undoubtedly rely on e-noses and biomarkers to be able to diagnose a vast range of conditions in a rapid, effective, non-invasive, painless, and low-cost manner.

### 4.4. Pulmonary Diseases

The gaseous exchange between blood and air that occurs in the interior of the alveoli allows for the exchange of a vast number of compounds from the interior of the organism to the exterior, and also from the exterior world to the interior of the body. Because of this exchange, the exhaled air can be a valuable source of information regarding the health state of the body, as already mentioned. Pulmonary diseases like lung cancer, asthma, cystic fibrosis, and tuberculosis are examples of pathologies traceable through the compounds existent in the breath and, specifically, ammonia [118,119].

Lung cancer is one of the most dangerous pulmonary diseases and its diagnosis has been explored regarding the suitability of ammonia as a biomarker. An e-nose used to detect ammonia in the breath of lung cancer patients was developed by Chen et al. (2020). To achieve this, exhaled air samples of 108 subjects were collected and analyzed with an exclusively developed e-nose, which consisted of an array of chemoresistive graphene-oxide-based sensors whose resistance variation was assessed when exposed to real samples of breath. Overall, the authors were able to successfully identify both groups, i.e., healthy individuals and lung cancer patients, with sensitivity and specificity levels of 95.8% and 96.0%, respectively [120].

Asthma is another pulmonary condition that figures among the most common pathologies worldwide. Its diagnosis and monitoring have been linked to the concentration levels of ammonia in breath, as explored by Hunt et al. (2002). By condensing and later analyzing breath from healthy individuals and asthmatic patients, the authors could notice different behaviors in terms of ammonia concentration. The levels of ammonia in the exhaled air condensate collected from healthy volunteers proved to be considerably higher (around 327 microM) than the levels registered for asthmatic patients (30 microM) [121].

Cystic fibrosis is another example of the utility of ammonia as a biomarker for purposes of diagnosis. With that in mind, Newport et al. (2009) collected and analyzed the exhaled air condensate of 20 volunteers (10 healthy volunteers and 10 patients with cystic fibrosis) concerning ammonia levels. The developed system allowed the authors to realize that the ammonia concentrations were considerably lower in the breath of cystic fibrosis patients when compared with the breath of healthy individuals [122].

Biomarkers of tuberculosis, a health condition provoked by the presence of the bacterium *Mycobacterium tuberculosis* and consequent infection, have been deeply explored for purposes of diagnosis [123]. The utilization of analytes and, specifically, of ammonia to diagnose tuberculosis is due to the metabolic process of the bacterium itself, which often releases several volatile compounds as a result of those processes. Reference [124] precisely explored this fact. The authors employed an e-nose based on gas chromatography and surface acoustic wave sensors to study the metabolic emissions of bacteria. During this laboratory study, ammonia and several other volatile organic compounds were identified as being emitted by the growth of the bacteria, making these compounds especially relevant for the field of biomarkers [124].

As reviewed, the potential of e-noses for the detection of ammonia and consequent diagnosis of pulmonary diseases is evident. The achieved results proved to be promising and useful to modern medicine.

Ammonia has been largely studied due to its potential usefulness in the medical field. The presence of ammonia in body fluids like exhaled air, saliva, or urine can represent an open window to the interior of the human organism and provide multiple data concerning several types of pathologies, namely renal, hepatic, pulmonary, and gastric conditions. Sensor-array-based systems and e-noses generally play an important role in the detection, identification, and quantification of such an important volatile compound. There is no doubt that both fields of ammonia as a biomarker and e-noses for its assessment can represent a valuable addition to the medical community and positively complement the current methodologies of diagnosis to help create the medicine of the future.

## 5. Concluding Remarks

Firstly, it is confirmed that research on ammonia e-noses for environmental applications is relevant, as the relationship between this topic and other related areas is not proportional, indicating a lack of research in this area. Next, the literature was collected and studied to determine if there were existing sensors ready to be commercialized for ensuring ammonia safety in the workplace. Information on 46 sensors was gathered and then filtered to align with safety requirements in the workplace, leaving only 18 viable options. These sensors were then compared to determine the best method for developing an ammonia sensor suitable for this context. The most promising method identified for developing an electronic ammonia sensor was a metal oxide sensor with a PANI/TiO_2_ film, grown at low temperatures. However, other PANI/TiO_2_ films that were considered yielded significantly inferior results compared to the best ones, raising concerns about the consistency of results. Thus, there were insufficient results to ensure the development of a commercially viable ammonia sensor for workplace safety, highlighting the need for further research. It is recommended that additional tests be conducted on the most successful sensor under different conditions to test its reproducibility and stability. These literature results suggest that the development of e-nose systems is advised to obtain more robust and accurate sensors in complex media. Additionally, the monitoring of ammonia traces in biological fluids serves as a versatile diagnostic tool, offering invaluable insights into the health status of multiple organs, like the liver, kidneys, stomach, and lungs. By leveraging sensors capable of detecting trace concentrations of ammonia (in the range of ppb_v_), healthcare professionals can diagnose and manage a diverse array of pathological conditions effectively. Fluctuations in ammonia levels serve as diagnostic markers for conditions such as hepatic impairment, renal impairment, gastric impairment, and pulmonary diseases. Examples include chronic liver disease (CLD), cirrhosis, chronic kidney disease (CKD), gastric cancer, lung cancer, asthma, cystic fibrosis, and tuberculosis. This proactive approach not only facilitates early intervention but also enhances patient care in the realm of clinical medicine, thereby improving patient outcomes in clinical medicine.

Finally, it should be added that the development of e-noses for detecting ammonia at low concentrations in complex environments bypasses the development of sensors using knowledge of the materials that allow for the detection of ammonia, the development of other sensors to detect other molecules present in the environment, the use of solutions that guarantee the increasing sensor lifespan, and using appropriate mathematical methods to make the devices reliable and maintain quality assurance and pass quality control if these sensors become commercial. Here, one should also reinforce that the biggest challenges can be described as finding reliable and long-lifetime sensors in the array of the e-nose.

## Figures and Tables

**Figure 1 sensors-24-03152-f001:**
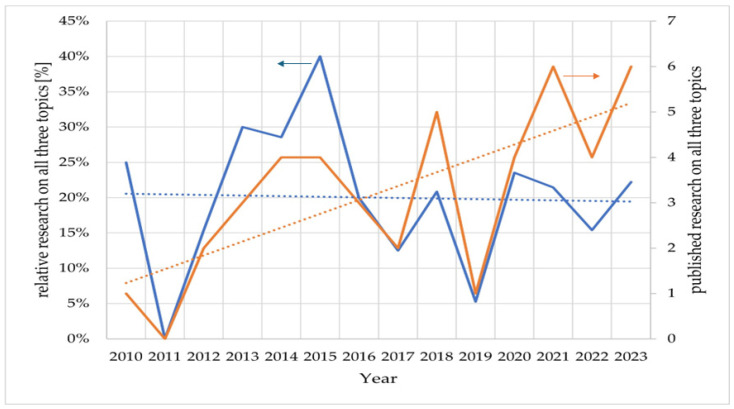
Literature research on “ammonia AND e-nose AND environment” relative to “ammonia AND e-nose” (blue). Absolute literature research on “ammonia AND e-nose AND environment” (orange). The blue and orange dotted lines are the linear fit of the relative research and the published research on those topics, respectively.

**Figure 2 sensors-24-03152-f002:**
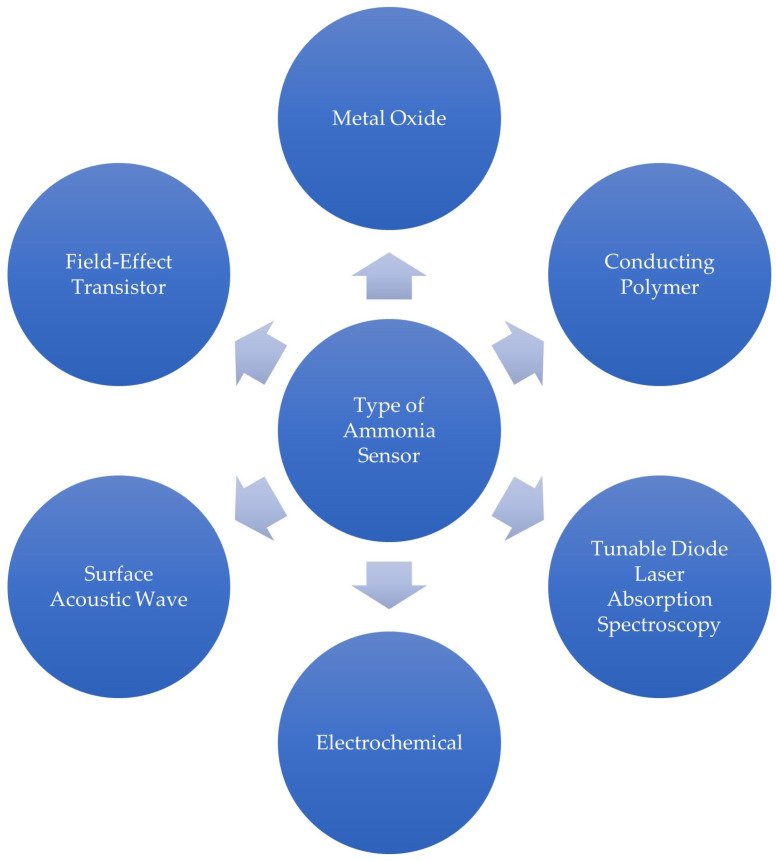
Types of ammonia sensors based on [25].

**Figure 3 sensors-24-03152-f003:**
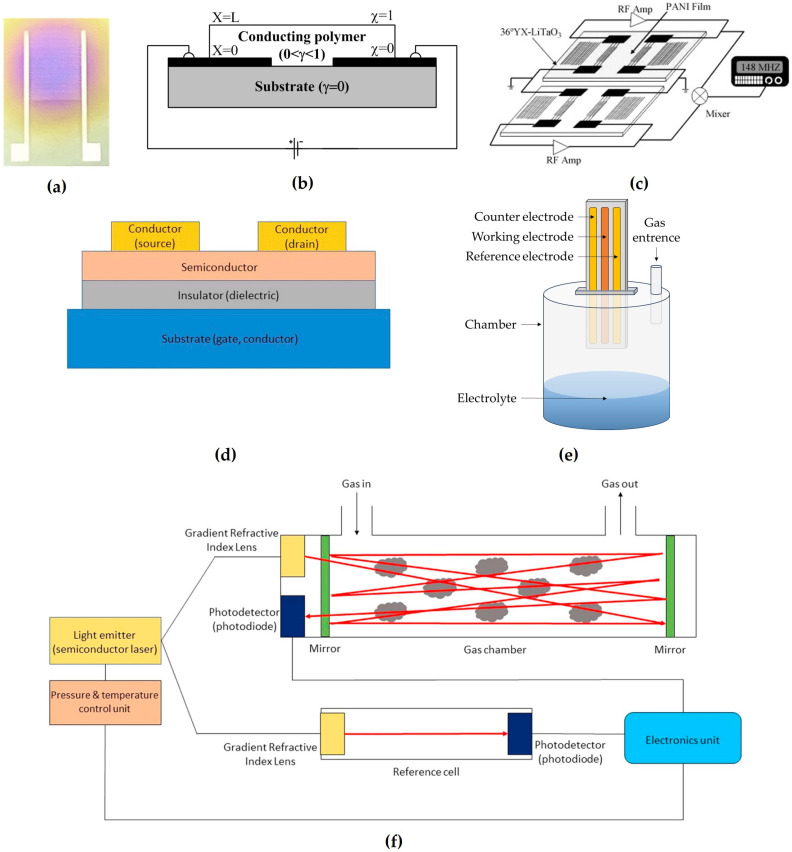
Layouts of different sensors used for ammonia gas detection: (**a**) Metal oxide reprinted from Ref. [39]; (**b**) conducting polymers reprinted from Ref. [29]; (**c**) SAW reprinted from Ref. [40]; (**d**) FET reprinted from Ref. [25]; (**e**) electrochemical; and (**f**) TDLAS reprinted from Ref. [25].

**Figure 4 sensors-24-03152-f004:**
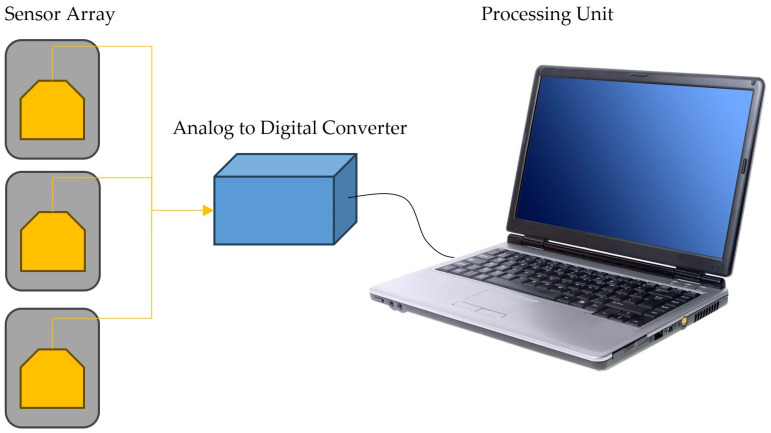
Schematic representation of an e-nose system, which consists of an array of sensors and one or more measuring system(s) with an analog-to-digital converter connected to a computer or a cellular phone for data treatment and data comparison.

**Figure 5 sensors-24-03152-f005:**
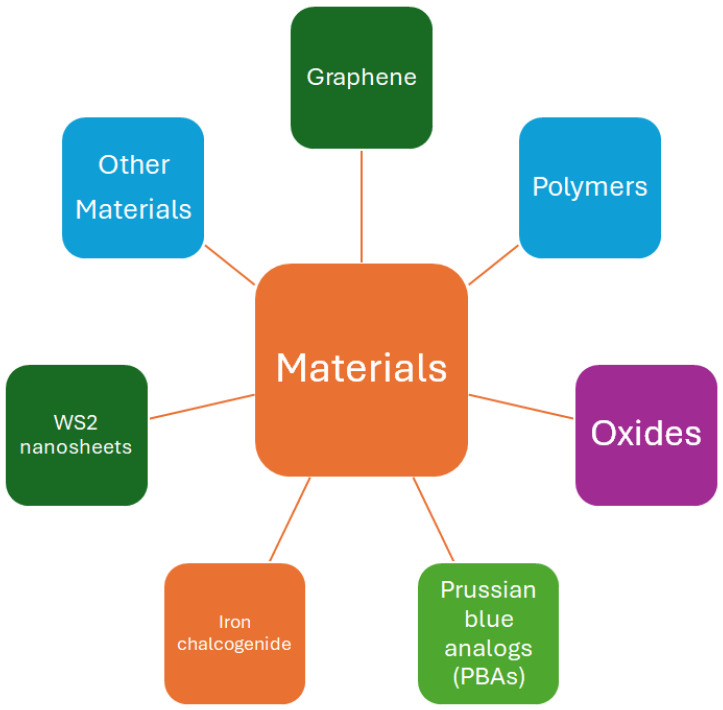
Materials used in the different types of ammonia sensors.

## Data Availability

No new data were created or analyzed in this study. Data sharing is not applicable to this article.

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
