# Peer review of "Ammonia Detection by Electronic Noses for a Safer Work Environment"

_sensors, 2024, doi:10.3390/s24103152_

Round 1
Reviewer 1 Report
Comments and Suggestions for Authors
The paper give a review on ammonia detection through artificial noses. The topic is interesting. But the content needs to be improved before final acceptance. Some suggestions is as follows:
1. In the introduction part. The sentences from line60-85 is too long. I suggests that the content should be only 5-7 lines. The figure 1 is not easy to understand. The paper of ammonia on e-nose and e-nose plus environment seems to be only 1% of that on environment. This only proves that the topic is not important for ammonia and environment field. I suggest remove this figures. Only use few sentences and figure 2.
2. In figure 3, it should be type of ammonia sensor instead of type of sensor.
3. the title use artificial noses and the content use e nose, I suggest the title should be changed. In the introduction part. No any information of e nose is given. The definition, history, component of e nose shall be given in the introduction.
4. The metal oxide gas sensors are listed, but there is not any description on the sensing performance of gas sensors. The performance including sensisitivity, response time, recovery time, selectivity should be added. The definition of these parameter should be given as well.
Comments on the Quality of English LanguageEnglish can be improved.
Author Response
The paper give a review on ammonia detection through artificial noses. The topic is interesting. But the content needs to be improved before final acceptance. Some suggestions is as follows:
Dear reviewer,
We want to thank you for your suggestions and fair evaluation of our work. Evidently, we properly addressed all the issues and hope this final version of the manuscript meets your quality standards.
- In the introduction part. The sentences from line60-85 is too long. I suggests that the content should be only 5-7 lines. The figure 1 is not easy to understand. The paper of ammonia on e-nose and e-nose plus environment seems to be only 1% of that on environment. This only proves that the topic is not important for ammonia and environment field. I suggest remove this figures. Only use few sentences and figure 2.
Answer: Thank you for this suggestion. The text was reduced, and Figure 1 was removed.
- In figure 3, it should be type of ammonia sensor instead of type of sensor.
Answer: Thank you for this suggestion. The figure was changed.
- the title use artificial noses and the content use e nose, I suggest the title should be changed.
Answer: We followed your suggestion and updated the title.
- In the introduction part. No any information of e nose is given. The definition, history, component of e nose shall be given in the introduction.
Answer: We thank you for this suggestion which we consider also necessary. A paragraph was included in the introduction.
- The metal oxide gas sensors are listed, but there is not any description on the sensing performance of gas sensors. The performance including sensisitivity, response time, recovery time, selectivity should be added. The definition of these parameter should be given as well.
Answer: Thank you for your suggestions. The sensing performance of the metal oxides is in Table 1 and the parameters associated with the sensor performance were defined in the manuscript.
Reviewer 2 Report
Comments and Suggestions for Authors
The review that authors submitted is not complete, Lot of different pounts have to be covered:
1. No sensing mechanisms of different materials liek oxides, polymers etc.., neither sensing mechanisms of different tehcnologies like FETS or Opticla etc.. were discussed with appropriate figuers and their corresponding permissions. It is mandatory to add this information.
2. Also authors didnot include the design and manufacturing process of e-noses or ammonia sensors with respective schematic diagrams.
3. QA and QC analysis fo the enoses and theor comparisonw ith practicla pplications has to be included too.
4. Novelty fo the work is missing, whihc has to be highlighted in the introduction section.
5. Future works and challenges also to be added in the final section.
Comments on the Quality of English LanguageMinor english corrections all over the manuscript is necessary.
Author Response
The review that authors submitted is not complete, Lot of different pounts have to be covered:
Dear reviewer,
We regret that you found our work incomplete and thank you for all the pertinent suggestions. We believe that, by addressing all the issues, the final version of the manuscript is considerably better and we hope it meets your quality standards.
- No sensing mechanisms of different materials liek oxides, polymers etc.., neither sensing mechanisms of different tehcnologies like FETS or Opticla etc.. were discussed with appropriate figuers and their corresponding permissions. It is mandatory to add this information.
Answer: Thank you for your suggestion. The information about the sensing mechanisms as well as the measuring approaches were added in the manuscript.
- Also authors didnot include the design and manufacturing process of e-noses or ammonia sensors with respective schematic diagrams.
Answer: Thank you very much for your suggestion. The schematic diagram for an e-nose was included in the manuscript.
- QA and QC analysis fo the enoses and theor comparisonw ith practicla pplications has to be included too.
Answer: We agree with your question. Although quality assurance (QA) and quality control (QC) are quite important issues in the case of commercial sensors, these subjects are not too much referred to in developing sensors in scientific research and we did not find information about these subjects in the available literature. However, in the conclusions, we added a paragraph referring to these subjects. Thank you very much for your question.
- Novelty fo the work is missing, whihc has to be highlighted in the introduction section.
Answer: Thank you for your very interesting suggestion. The novelty and goal of this work are addressed in a dedicated paragraph of the introduction, namely, between lines 49 – 58: “Despite all studies already developed and content previously available, the monitoring of ammonia gas is still not normalized mainly due to the unavailability of appropriate commercial sensors. A lack of standard procedures used to construct ammonia sensors is a major contemporary challenge in assessing ammonia. Additionally, there is a lack of appropriate commercial sensors to detect ammonia, evaluate work conditions for all kinds of workplaces, and prevent accurate identification of main health impacts on human organisms in chronic exposure scenarios. Our review can help promote the most promising methods for the development of ammonia sensors. We present the latest options in ammonia detection aimed at determining which sensors can accurately detect levels of ammonia that satisfy those previously mentioned regulations, i.e., up to 20 ppmv.”
- Future works and challenges also to be added in the final section.
Answer: A paragraph was included at the end of the manuscript. Thank you very much for your suggestion.
Reviewer 3 Report
Comments and Suggestions for Authors
The Manuscript titled “Ammonia Detection by Artificial Noses for a Safer Work Environment" is well-written and comprehensively reviews the particular subject. The article can be published in Sensors, subject to revisions as follows:
1. Lines 50-59 Please cite reference
2. Line 111-113 Elaborate disadvantages
3. Line 128 - Is it most FET or MOSFET?
4. Line 134 - Table 1 presents the sensors with a response time under 1.3 minutes at a concentration of (25±15) ppm of NH3…..My point is the highest C (ppm) is 50. So, revise the 25±15. Also, in table 1 representation, add footnotes and explain abbreviation of “C”.
5. Lines 147-148 The film ……….temperatures). Explain the point clearly. Also, correct temperatures[45] to temperatures [45] and wheresoever throughout the manuscript.
6. Line 157 - Correct the sentence “showed the best results[35],,” to showed the best results [35],
7. Line 216 - In section 3 add working principle of E-nose and represent it diagrammatically
8. Line 240 - Explain why is the data processing unit considered one of the most important components of e-noses?
9. Line 243 – Mention important algorithms which can be employed for the data interpretation of enoses. You can refer to the article “Applications of Electronic Nose Coupled with Statistical and Intelligent Pattern Recognition Techniques for Monitoring Tea Quality: A Review” by Kaushal et al. 2022.
10. Lines 244-248 Add reference
11. Line 245- What are the distinguishing characteristics of sensors based on metal oxides versus polymers? How do these characteristics align with the requirements for a suitable response time?
12. Line 253- Section 4 Other applications of e-noses of ammonia – revise the heading.
13. Line 289 CLD abbreviation is already explained in line 279. Always explain for the first-time use.
14. Line 289-292 Rewrite again
15. Line 349 - Abbreviated form of CKD is repeated again.
16. In Acknowledgement and conflict of interest section authors should change the copied statement
Comments on the Quality of English LanguageLanguage quality of manuscript is good and presently adequate
Author Response
The Manuscript titled “Ammonia Detection by Artificial Noses for a Safer Work Environment" is well-written and comprehensively reviews the particular subject. The article can be published in Sensors, subject to revisions as follows:
Dear reviewer,
We want to thank you for your fair evaluation of our work and all the useful comments. Please see below all the improvements of the paper. We hope that this final version meets your expectations.
- Lines 50-59 Please cite reference
Answer: A reference was included in this place.
2.Line 111-113 Elaborate disadvantages
Answer: The disadvantages were elaborated. Thank you.
- Line 128 - Is it most FET or MOSFET?
Answer: The writing “most FET” is correct since it intends to mean “the majority of FET (field-effect technologies)”.
- Line 134 - Table 1 presents the sensors with a response time under 1.3 minutes at a concentration of (25±15) ppm of NH3…..My point is the highest C (ppm) is 50. So, revise the 25±15. Also, in table 1 representation, add footnotes and explain abbreviation of “C”.
Answer: The interval was revised, the table caption was improved, and C was defined.
- Lines 147-148 The film ……….temperatures). Explain the point clearly. Also, correct temperatures[45] to temperatures [45] and wheresoever throughout the manuscript.
Answer: Corrected. Thank you
- Line 157 - Correct the sentence “showed the best results[35],,” to showed the best results [35],
Answer: Thank you for alerting us, corrections were implemented.
- Line 216 - In section 3 add working principle of E-nose and represent it diagrammatically
Answer: Thank you for your suggestion, the schematic representation of an e-nose was added to the manuscript.
- Line 240 - Explain why is the data processing unit considered one of the most important components of e-noses?
Answer: Thank you for your question. An explanation of data processing was included.
- Line 243 – Mention important algorithms which can be employed for the data interpretation of enoses. You can refer to the article “Applications of Electronic Nose Coupled with Statistical and Intelligent Pattern Recognition Techniques for Monitoring Tea Quality: A Review” by Kaushal et al. 2022.
Answer: The reference was added. Thank you.
- Lines 244-248 Add reference
Answer: We do not have a reference since it is our suggestion to solve the problem of the sensors do not recover after several measurements.
- Line 245- What are the distinguishing characteristics of sensors based on metal oxides versus polymers? How do these characteristics align with the requirements for a suitable response time?
Answer: Thank you for your question. Metal oxides and polymers are both materials with suitable response times and the most adequate sensors to be submitted to temperature or vacuum.
- Line 253- Section 4 Other applications of e-noses of ammonia – revise the heading.
Answer: The heading was revised, thank you.
- Line 289 CLD abbreviation is already explained in line 279. Always explain for the first-time use.
Answer: We apologize for this issue and, evidently, we corrected it.
- Line 289-292 Rewrite again
Answer: This paragraph was rewritten to improve the clarity, thank you.
- Line 349 - Abbreviated form of CKD is repeated again.
Answer: Thank you for alerting us. Corrections were implemented.
- In Acknowledgement and conflict of interest section authors should change the copied statement
Answer: Thank you for alerting us. Corrections were implemented.
Round 2
Reviewer 1 Report
Comments and Suggestions for Authors
I think the present form is fine
Comments on the Quality of English LanguageEnglish is OK.
Author Response
Dear reviewer, thank you for accepting our work in its present form and for all your helpful comments.
Best regards,
Maria Raposo
Reviewer 2 Report
Comments and Suggestions for Authors
All the comments raised were replied by authors. Manuscript cna be accepted in the current form.
Comments on the Quality of English LanguageAll the comments raised were replied by authors. Manuscript can be accepted in the current form.
Author Response
Dear Reviewer, thank you for accepting our work in its present form and for all your helpful comments.
Best regards,
Maria Raposo